# S3: Supervised Self-supervised Learning under Label Noise

## Abstract

Despite the large progress in supervised learning with Neural Networks, there are significant challenges in obtaining high-quality, large-scale and accurately labeled datasets. In this context, in this paper we address the problem of classification in the presence of label noise and more specifically, both close-set and open-set label noise, that is when the true label of a sample may, or may not belong to the set of the given labels. In the heart of our method is a sample selection mechanism that relies on the consistency between the annotated label of a sample and the distribution of the labels in its neighborhood in the feature space; a relabeling mechanism that relies on the confidence of the classifier across subsequent iterations; and a training strategy that trains the encoder both with a self-consistency loss and the classifier-encoder with the cross-entropy loss on the selected samples alone. Without bells and whistles, such as co-training so as to reduce the self-confirmation bias, and with robustness with respect to settings of its few hyper-parameters, our method significantly surpasses previous methods on both CIFAR10/CIFAR100 with artificial noise and real-world noisy datasets such as WebVision and ANIMAL-10N.

## 1 Introduction

It is now commonly accepted that supervised learning with deep neural networks can provide excellent solutions for a wide range of problems, so long as there is sufficient availability of labeled training data and computational resources. However, these results have been mostly obtained using well-curated datasets in which the classes are balanced and the labels are of high quality. In the real-world, it is often costly to obtain high quality labels especially for large-scale datasets. A common approach is to use semi-automatic methods to obtain the labels (e.g. "webly-labeled" images where the images and labels are obtained by web-crawling). While such methods can greatly reduce the time and cost of manual labeling, they also lead to low quality noisy labels.

To deal with noisy labels, earlier approaches tried to improve the robustness of the model using robust loss functions (Ghosh et al., 2017; Zhang & Sabuncu, 2018; Wang et al., 2019) or robust regularizations (Srivastava et al., 2014; Zhang et al., 2017; Pereyra et al., 2017). Goldberger & Ben-Reuven (2016) tried to model the noise transition matrix between classes while Han et al. (2019); Patrini et al. (2017); Hendrycks et al. (2018) proposed to correct the losses of noisy samples. More recently, sample selection methods became perhaps the dominant paradigm for learning with noisy labels. Most of the recent sample selection methods do so, by relying on the predictions of the model classifier, for example on the per-sample loss (Arazo et al., 2019; Li et al., 2020a) or model prediction (Song et al., 2019; Malach & Shalev-Shwartz, 2017). By separating clean samples and noisy samples and subsequently performing supervised training on the clean set, or semi-supervised training on both, sample selection methods achieved the state-of-the-art results in synthetic and real-world noisy datasets.

However, there are three main issues with current sample selection methods. Firstly, the sample selection will be inevitably biased if the models (classifier and feature extractor) are trained with noisy labels – this is immediately apparent in the case that the sample selection is based on the loss of the classifier itself (Arazo et al., 2019; Li et al., 2020a; Yu et al., 2019; Han et al., 2018). Second, in supervised classification problems, noisy samples usually come from two main categories: closed-set noise where the true labels belong to one of the given classes (Set B in Fig. 1) and open-set noise

where the true labels do not belong to the set of labels of the classification problem (Set C in Fig. 1). Most of the works in the literature, including works that estimate the probabilities of label-exchange between pairs of classes (Goldberger & Ben-Reuven, 2016; Patrini et al., 2017), that do relabeling based on the model's predictions (Song et al., 2019; Han et al., 2019) or works that adopt semi-supervised approaches (Li et al., 2020a; Ortego et al., 2021) deal with the former and not directly address the latter. However, this is a considerable source of noise in real world scenarios, e.g., when training from web-crawled data, where there is less control over the collection of the dataset. Crucially, those works (implicitly or explicitly) relabel all samples and do training based on the new labels of all samples. Those are bound to be wrong for all samples in set C, but also are bound to be wrong for several samples of A and B that are (implicitly or explicitly) relabeled in the early stages of training. For the latter reason, those works do not work well even under heavy close-set noise. Finally, current approaches usually require extensive hyperparameters tuning, often even on a per-dataset basis – this is unrealistic in scenarios where there is little knowledge about the types of noise. This is partly due to the complexity of the applied semi-supervised learning method, and partly because of the complicated methods that are employed, such as model pretraining (Zheltonozhskii et al., 2021) and model cotraining (Han et al., 2018; Yu et al., 2019; Li et al., 2020a), so as to deal with self-confirmation bias.

In this paper, we address the problem of training under different types of noise with a simple method–**namely Supervised, Self-Supervised learning (S3)**, with two major components that are clearly separated: a selection/relabelling mechanism that selects/relabels samples so as to construct a clean and a noisy set (Section 3.3), and a training framework that aims at learning a strong feature extractor $f$ and a classification head $g$ [Fig. 2] from both noisy and clean samples (Section 3.4). In the training stage we use off-the-shelf learners and a) train the feature extractor and the classification head using a classical **Supervised cross-entropy loss** applied only on the clean samples – this avoids treating samples identified as noisy as if they belonged to one of the given classes as most methods (Arazo et al., 2019; Li et al., 2020a; Ortego et al., 2021; Wu et al., 2021) implictly or explicitly do; and b) train the feature extractor using a **Self-Supervised loss**, namely the consistency loss between the representations of augmented version of the sample (as in (Chen & He, 2021)) applied on all samples – this avoids false-negatives that are inherent in contrastive-learning with instance discrimination and different to works that apply the consistency on the label predictions that are unreliable at the first iterations or in the presence of open-set noise. The noisy sample selection mechanism relies on a measure of confidence what we define using the ground truth label of the sample in question and an estimate of distribution of the labels of its neighbours – in order to deal with noisy samples, we adopt a scheme in which the distribution is calculated based both on the ground truth labels and on consistently (over subsequent iterations) confident estimates of the labels. Our method is embedded into a standard MixUp and data augmentation framework, and without bells and whistles, such as co-training of multiple models, it achieves state-of-the-art results in both synthetic and realistic noise patterns in CIFAR10, CIFAR100, ANIMAL-10N, Clothing1M and WebVision datasets.

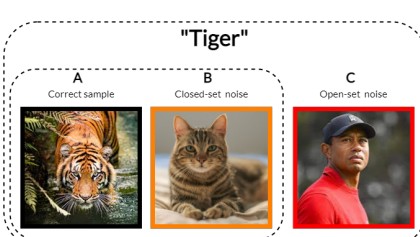

Figure 1: Different 'tigers' in an animal dataset.

## 2    RELATED WORK

**Leaning with noisy data by sample selection**    Some works focused on sample selection to filter out noisy samples. Jiang et al. (2018) introduced a pretrained mentor network to guide the selection of a student network. Song et al. (2019) evaluates the per-sample losses and identify as clean the top $r\%$ of the samples – the precise ratio $r\%$ depends is either predefined, or is an estimate of the noise level in the specific dataset. Arazo et al. (2019) proposed to model per-sample losses with a Beta Mixture Model (BMM) and split the dataset according to which of the components of the mixture each sample belongs. In a very similar approach, Li et al. (2020a) extended upon Arazo et al. (2019) by introducing semi-supervised learning to fully utilize the dataset. Related to our work, Bahri et al. (2020); Ortego et al. (2021); Wang et al. (2018) also utilized the feature space for sample selection. Bahri et al. (2020) applied KNN for sample selection for closed-set noisy dataset while Ortego et al.

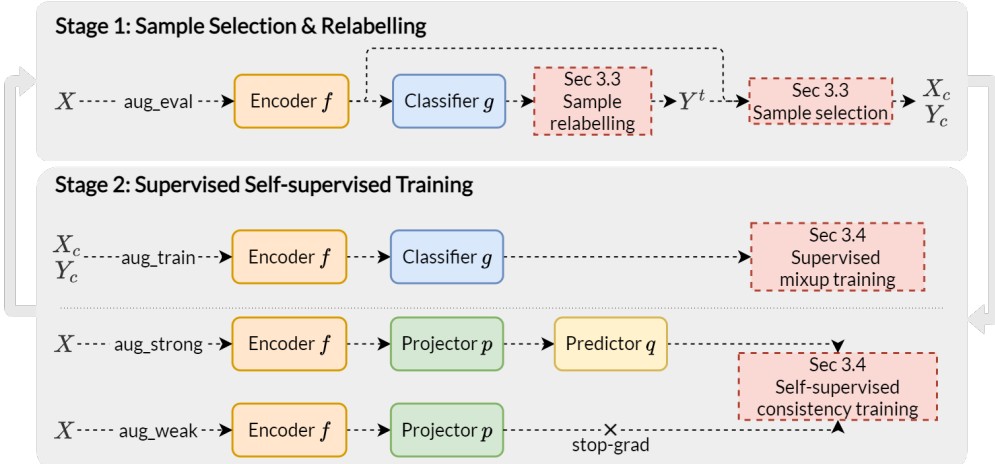

Figure 2: S3 consists of two iterative stages: sample selection&relabelling and supervised self-supervised training.

(2021) further proposed to relabel samples based on the KNN voting. Wang et al. (2018) proposed to reweight samples based on its probability of being outliers in open-set noisy dataset.

**Self-supervised learning** Self-supervised methods attempt to learn good representation without human annotations. In the recent years, the dominant method is contrastive learning with instance discrimination task. MoCo (He et al., 2020) is an important baseline for current contrastive learning methods, which reuses the memory bank since samples in a single mini-batch may lead to insufficient negative pairs, and proposes a momentum encoder to update the memory bank in real-time to avoid outdated data representation. SimCLR (Chen et al., 2020) is another important baseline which found that setting mini-batch size to be large enough can eliminate the need for memory bank. More recently, SimSiam (Chen & He, 2021) and BYOL (Grill et al., 2020) proposed a non-contrastive learning framework which enforce the perturbation consitency between different views, and avoid mode collapse by applying stop-gradient and an extra predictor between two representation vectors.

## 3 METHOD

### 3.1 PROBLEM FORMULATION

Let us denote with $\boldsymbol{X} = \{\boldsymbol{x}_i\}_{i=1}^N, \boldsymbol{x}_i \in R^d$, a training set with the corresponding one-hot vector labels $\boldsymbol{Y} = \{\boldsymbol{y}_i\}_{i=1}^N, \boldsymbol{y}_i \in \{0,1\}^K$, where $K$ is the number of classes and $N$ is the number of samples. For convenience, let us also denote the index where the one-hot vector $\boldsymbol{y}_i$ is one as the label $l_i \in \{0, ..., K\}$. Finally, let us denote the true labels with $\boldsymbol{Y}' = \{\boldsymbol{y}_i'\}_{i=1}^N$. Clearly, for an open-set noisy label it is the case that $\boldsymbol{y}_i' \neq \boldsymbol{y}_i, \boldsymbol{y}_i' \notin \{0,1\}^K$, while for closed-set noisy samples $\boldsymbol{y}_i' \neq \boldsymbol{y}_i, \boldsymbol{y}_i' \in \{0,1\}^K$.

### 3.2 OVERVIEW OF PROPOSED METHOD

Aiming to deal with potential concurrence of both open-set and close-set noise, we view the classification network as an encoder $f$ that extracts a feature representation and a classification head $g$ that deals with the classification problem in question. The proposed method, named Supervised Self-Supervised(S3) learning, attempts to decouple their training so as to deal with possible noise in the labels, by adopting a two stage, iterative scheme, as outlined in Fig. 2.

In the first stage, we utilize a novel sample selection and a novel relabeling mechanism (top block in Fig. 2) that prepares the set based on which the classifier $g$ should be trained in Stage 2. The selection mechanism is based on the assumption of smoothness of labels in the feature space, and more specifically, on a consistency measure that we defined based on the annotated label of the sample

in question and the distribution of the labels in its neighborhood in the feature space. Relabeling is performed on samples for which the classifier gives confident predictions consistently across subsequent iterations. Clearly, the mechanism relies on the quality of the features extracted by the encoder $f$ and should reject samples whose true labels are not in the class set (open-set noise). This stage is explained in Section 3.3. In the second stage (bottom block in Fig. 2), training is performed with two objectives/losses. First, a cross entropy loss on the output of the classifier $g$, (i.e., on $g(f(.))$) on the samples selected in Stage 1, that updates both the encoder $f$ and the classifier head $g$. Second, a self-supervision loss that enforces consistency between the representations of different augmentations of the same sample, and which utilizes all samples, that is both noisy and clean – this updates the encoder $f$ and helps learning a strong feature space on which the selection mechanism of Stage 1 can rely. By contrast to other methods that, either by using a noise transition matrix or in their semi-supervised scheme, implicitly relabel all samples (e.g. DivideMix (Li et al., 2020a), MOIT (Ortego et al., 2021)) and use the new labels to learn, in our method the labels in the noisy set are not used at all. This stage is explained in Section 3.4.

### 3.3 SAMPLE SELECTION & RELABELLING

The sample selection and relabeling mechanism are designed so as to construct a clean subset that has as many correctly labelled samples as possible.

**Clean sample selection by balanced neighboring voting**  Let us denote the similarity between the representations $f_i$ and $f_j$ of any two samples $x_i$ and $x_j$ by $s_{ij}, i, j = 1, ..., N$. In our implementation we used the cosine similarity, that is, $s_{ij} \triangleq \frac{f_i^T f_j}{\|f_i\|_2 \|f_j\|_2}$. Let us also denote by $N_i$ the index set of the $k$ nearest neighbors of sample $x_i$ in $X$ based on the calculated similarity. Then, for each sample $x_i$, we calculate the balanced label distribution $p_i \in R^K$ in its neighborhood in the feature space, as the normalized sum of its neighbors' labels. Note, that we used at each epoch $t$, we use the labels $y_n^t$ (Eq. 2) that a relabeling mechanism provides. More specifically,

$$p_i = \pi^{-1} p_i', \tag{1}$$

where $p_i' = \frac{1}{k} \sum_{n \in N_i} y_n^t$ and $\pi = \sum_{i=1}^N y_i$. With a slight abuse of notation, we denote by $\pi^{-1}$ the vector whose entries are the inverses of the entries of the vector $\pi$ of the class probabilities in the whole dataset – in this way we compensate for global class imbalances. Once the balanced distribution $p_i$ of the labels in the neighborhood of $x_i$ is estimated, we define a consistency measure $c_i$ as

$$c_i = \frac{p_i(l_i)}{\max_l p_i(l)}, \tag{2}$$

that is the ratio of the value of the distribution $p_i$ at the label $l_i$ divided by the value of its highest peak $\max_l p_i(l)$. Roughly speaking, a high consistency measure $c_i$ at a sample $i$ means that its neighbors agree with the given annotation $l_i$ – this indicates that $l_i$ is likely to be correct. By setting a threshold $\theta_s$ to $c_i$, a clean subset $(X_c, Y_c)$ can be extracted from the noisy dataset.

**Noisy sample relabelling by classifier thresholding**  The balanced distribution $p_i$ of the labels of the neighbors of a sample $i$ constructed by Eq. 1 is to some degree affected by the noisy labels of its neighbors. One option is to use the computed normalized nearest neighbor distribution $p_i$ to relabel the noisy samples – for example, relabel the sample $x_i$ as $\arg \max p_i$ (Ortego et al., 2021). However, this process will introduce self-confirmation bias as we relabel and select samples by relying in both cases on the feature space.

In this paper, we propose to decouple sample selection and sample relabeling. More specifically, given that the classifier is trained in subsequent iterations with relatively clean samples (selected by the mechanism described above), we propose to used its predictions to identify samples for which it has high confidence in its prediction, and relabel them if the confident prediction does not agree with the annotated label. More specifically, let us denote by $z_i^t \triangleq g(f(x_i))$ the prediction at epoch $t$ for sample $x_i$. By keeping track of the most recent $L$ predictions, we calculate

$$q_i = \frac{1}{L} \sum_{t'=t-L}^{t} z_i^{t'}, \tag{3}$$

where $\boldsymbol{q}_i$ is the average prediction for the sample $\boldsymbol{x}_i$. We then modify all the labels $l_i, i = 1, ..., N$ by thresholding $\boldsymbol{q}_i$ at $t$-th epoch as:

$$l_i^t = I(\max_l \boldsymbol{q}_i(l) > \theta_r) * \arg\max_l \boldsymbol{q}_i(l) + (1 - I(\max_l \boldsymbol{q}_i(l) > \theta_r)) * l_i \qquad (4)$$

Please note, that similarly to Section 3.1, we denote the one-hot label corresponding to $l_i^t$ as $\boldsymbol{y}_i^t$ – this will be used in Eq. 1. By setting a high $\theta_r$, a highly confident sample $\boldsymbol{x}_i$ will be relabeled – this can in turn further enhance the quality of sample selection. Note, that we avoid mis-relabelling open-set noise samples as those tend not to have highly confident average predictions.

## 3.4 SUPERVISED SELF-SUPERVISED TRAINING

For training, we have both a supervised learning loss on the clean subset (cross entropy) and a self-supervised learning loss (consistency). The latter makes no assumption at all (implicit or explicit) about the labels and therefore can be applied on all samples.

**Supervised mixup training of the encoder-classifier using the clean subset**   With two random samples $\boldsymbol{x}_1, \boldsymbol{y}_1$ and $\boldsymbol{x}_2, \boldsymbol{y}_2$ in the clean subset $(\boldsymbol{X}_c, \boldsymbol{Y}_c)$, a mixed new sample $\boldsymbol{x}_m, \boldsymbol{y}_m$ will be generated by Mixup method (Zhang et al., 2017) as:

$$\lambda \sim Beta(\alpha, \alpha), \lambda' = \max(\lambda, 1 - \lambda), \boldsymbol{x}_m = \lambda'\boldsymbol{x}_1 + (1 - \lambda')\boldsymbol{x}_2, \boldsymbol{y}_m = \lambda'\boldsymbol{y}_1 + (1 - \lambda')\boldsymbol{y}_2 \quad (5)$$

We then apply the normal cross-entropy loss for the new virtual mixed sample:

$$L_{ce} = -\sum_{c=1}^K \boldsymbol{y}_m^c log \boldsymbol{z}_m^c \qquad (6)$$

where $\boldsymbol{z}_m = Softmax(g(f(\boldsymbol{x}_m)))$. Instead of direct training with samples from the clean subset, we expect that virtual samples generated by Mixup are further away from the dataset samples thus can alleviate the noise memorization effect (Zhang et al., 2017). This loss is back-propagated so as to update both the encoder $f$ and the classification head $g$.

**Self-supervised consistency training of the encoder using all samples**   To fully utilize all the samples, we applied a self-consistency loss motivated by recent non-contrastive self-supervised learning methods (Chen & He, 2021; Grill et al., 2020). With a projector head $p$ and prediction head $q$, we minimize the negative cosine similarity between two different augmented views from the same sample $x_i$. Denoting the two output vectors as $\boldsymbol{q}_i \triangleq \boldsymbol{q}(p(f(\boldsymbol{x}_{i1})))$ and $\boldsymbol{p}_i \triangleq \boldsymbol{p}(f(\boldsymbol{x}_{i2}))$:

$$L_{sc} = -\frac{\boldsymbol{q}_i^T \boldsymbol{p}_i}{\|\boldsymbol{q}_i\|_2 \|\boldsymbol{p}_i\|_2}, \qquad (7)$$

where $\boldsymbol{x}_{i1}, \boldsymbol{x}_{i2}$ denotes two different augmented views. $L_{sc}$ bears similarity to the commonly used consistency regularization in semi-supervised learning methods, however, the consistency is enforced between the projected features rather than the predictions. This allows us to utilize open-set noise also for training whose true labels are not in the label set. Also, we applied gradient stopping and an extra predictor so as to avoid mode collapse. This loss is back-propagated so as to update the encoder $f$ and contributes to learning a strong feature space from both clean and noisy samples. We also investigated on the use of the L2 loss as distance metric – details can be found in Appendix D.

**Data augmentations**   Strong augmentations, such as Cubuk et al. (2020; 2019), have shown to be effective in both supervised and semi-supervised learning (Berthelot et al., 2019; Sohn et al., 2020) and recently, Nishi et al. (2021) validated the benefits of strong augmentations within the DivideMix (Li et al., 2020a) framework. In this work, we define and use three types of augmentations: the original image itself (augmentation type 'none') is used for testing, random cropping and horizontal flipping (augmentation type 'weak'), and the augmentation policy proposed in Cubuk et al. (2019) (augmentation type 'strong'). In the model training phase, by default, we apply 'strong' augmentation for $\boldsymbol{x}_{i1}$, 'weak' augmentation for $\boldsymbol{x}_{i2}$ in $L_{sc}$, and 'strong' augmentation for $\boldsymbol{x}_1, \boldsymbol{x}_2$ in $L_{ce}$. In the sample selection and relabelling phase, we apply 'weak' augmentation so as to introduce more variance and alleviate the accumulation of error – this is in contrast to most works that do not apply augmentations at similar stages.

**Balanced sampler** In order to address possible class imbalances in the noise-agnostic dataset, we balanced the local distribution of class labels, with the inverse of the global class probabilities vector $\boldsymbol{\pi}$ during sample selection (Eq. 1). Similarly, the extracted clean subset $(\boldsymbol{X}_c, \boldsymbol{Y}_c)$ might also potentially suffer from class imbalances. To deal with this, we also use a balanced sampler for training with $L_{ce}$ by oversampling the minority class.

The overall training objective is to minimize a weighted sum of $L_{ce}$ and $L_{sc}$.

$$L = L_{ce} + wL_{sc} \tag{8}$$

while for all experiments, we fix $w = 1$.

## 4 EXPERIMENTS

### 4.1 OVERVIEW

In this section, we conduct extensive experiments on two standard benchmarks with artificial label noise, CIFAR-10 and CIFAR-100, and two real-world datasets, WebVision and ANIMAL-10N (see Appendix A for details). In Section 4.2.1, we conduct extensive ablation experiments to show the robustness of our method w.r.t its hyperparameters with different noise types, noise ratios and dataset. In Section 4.2.2, we conduct extensive ablation studies to validate the benefits of different modules in our method. In Section 4.2.2, we compared S3 with related works in each single stage. In Section 4.3 and 4.4, we compared with the state-of-the-art in synthetic noisy datasets and real-world noisy datasets. Implementation details can be found in Appendix B.

### 4.2 ABLATIONS STUDY

In this section, we conduct extensive ablation experiments to show the robustness w.r.t. the few hyperparameters with different noise types, noise ratios and dataset. In addition, to clearly show the contributions of different components and choices in S3 we perform experiments where we replace our training or sample selection/relabeling components with those from other methods in the literature.

### 4.2.1 ANALYSING SAMPLE SELECTION AND RELABELLING

We aim at building a framework which is robust in noise-agnostic dataset scenario and with minimal number of hyperparameters. In this section, we conduct extensive ablation experiments to show the robustness of the few hyperparameters with different noise types, noise ratios and dataset.

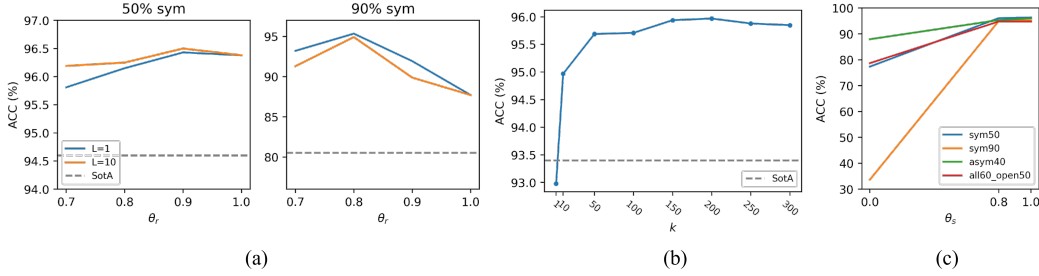

Figure 3: Classification accuracy with different $\theta_r$, $L$, $k$ and $\theta_s$ of synthetic CIFAR10 datasets. (a) $\theta_r = [0.7, 0.8, 0.9, 1]$, $L = [1, 10]$; (b) $k = [1, 10, 50, 100, 150, 200, 250, 300]$; (c) $\theta_s = [0, 0.8, 1.0]$.

$\theta_r$ **and** $L$ **in sample relabelling** The choice of $\theta_r$ and $L$ controls the sample relabelling quality and proportion. Roughly speaking, the lower the $\theta_r$ and $L$, the more samples will be relabeled in the training process, which also means that possibly more errors will be introduced. In Fig. 3(a) we reported performance with different combinations of $\theta_r$ and $L$ on the synthetic CIFAR10 noisy dataset. Genrally, our method achieved superior performance than the state-of-the-art with different $\theta_r$ and $L$. For example in CIFAR10 dataset with 90% sym noise, the lowest accuracy is 87.79% – this surpasses the state-of-the-art by $\sim$7%.

**Robustness w.r.t $k$ and $\theta_s$ in sample selection**    The number $k$ controls the neighborhood size in the sample selection phase. In Fig. 3(b), we report results with different $k$ for the CIFAR10 dataset with 40% asym noise. Except for too small $k$ which is more sensitive to the noisy samples, the performance is stable and consistently higher than the state-of-the-art. $\theta_s$ controls the number of selected samples, with $\theta_s = 0$ corresponding to no sample selection (i.e., all samples considered as clean subset). In Fig. 3(c), we report results with $\theta_s = [0, 0.8, 1.0]$. Removing sample selection leads to severe degradation especially in high noise ratio (90% symmetric noise), while a relatively high $\theta_s$ giving consistently high performance. We fixed $\theta_s = 1$ for experiments.

**Peformance of sample selection and relabelling**    At each iteration $t$, after relabelling and before sample selection, S3 creates a labeled set in which one can find correctly labelled samples (let's denote this subset by $A^t$), wrongly labeled samples that could be labeled with one of the given labels (subset $B^t$) and the out of set noise (subset $C^t$). Note, the correspondence with the sets $A, B$ and $C$ in the original labeled set depicted in Fig.1. After the sample selection, some will be selected and some not. Ideally, the relabelling mechanism will relabel all samples correctly (i.e., $A^t = A + B$) and, at every iteration, the selection mechanism will select only the correctly labeled samples (i.e., those in $A^t$). In order to show the performance of the selection mechanism, we report in Fig. 4(a) the F-score for the selection. In order to show the performance of the relabeling mechanism, we show in Fig. 4(b) the ratio of the correctly labeled samples (i.e., $A^t/(A+B+C)$. It is clear that the relabelling mechanism succeeds in increasing the size of correctly labeled samples ($A^t$) even in the case of heavy or out-of-set noise. It is also clear that in all cases the sample selection mechanism achieves high scores, even in the challenging cases of heavy noise.

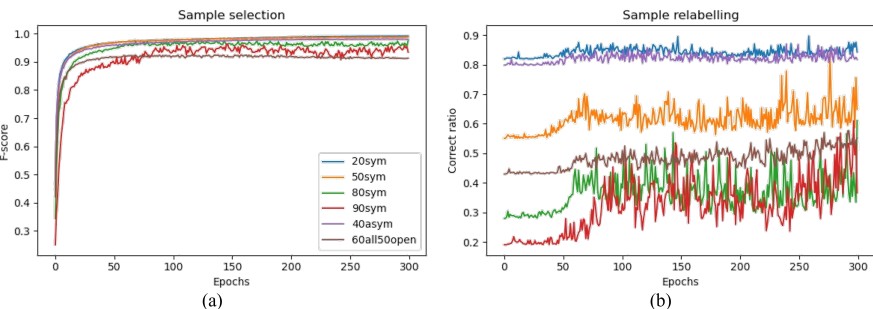

Figure 4: (a) F-score of the sample selection mechanism; (b) Ratio of samples with correct labels after sample relabelling.

### 4.2.2 TRAINING MODULES ABLATIONS

**Effects of self-consistency and Mixup**    In Table 1 we report the effect of self-consistency regularization and Mixup in S3 training stage. Removing Mixup decreases the performance, especially in high noise ratio and removing self-consistency also lead to degradation. Both help prevent memorization of wrong selections and to explore all samples so as to improve the model robustness.

| Dataset | 50% sym | 90% sym | 40% asym |
|---|---|---|---|
| S3 | **96.25** | **94.92** | **95.97** |
| w/o Mixup | 94.13 | 83.11 | 93.99 |
| w/o Self-consistency | 95.80 | 93.31 | 95.54 |

Table 1: Classification accuracy w/o Mixup and Self-consistency.

| Method | Clothing1M | 40% asym CIFAR10 |
|---|---|---|
| S3 | **74.91** | **95.97** |
| w/o balancing | 74.12 | 95.39 |

Table 2: Effect of balancing strategies.

**Effect of balancing strategies**    To alleviate possible dataset class imbalance, we propose two balancing strategies in sample selection and model training phase, respectively. In Table 2 we investigate its effect with different controlled noise in CIFAR10 and also a well-known real-world imbalanced noisy dataset Clothing1M.

### 4.2.3 ABLATIONS BY REPLACING COMPONENTS

To demonstrate the contributions of our choices, we substitute components of our method, with components of two state of the art methods, i.e., MOIT and DivideMix, and more specifically, their sample selection and training stages. The results are summarised in Table 3. It is clear that the semi-supervised training scheme that MOIT adopts, in which all samples are (implicitly) relabelled, and the DivideMix sample selection that is based on the classifier's prediction, perform significantly worse than ours, especially in the case of high noise ratios.

| Sample selection | Training | 50% sym | 90%sym | 40% asym | 60% all(50% open-set) |
|---|---|---|---|---|---|
| S3 | S3 | **96.30** | **95.20** | **96.0** | **94.81** |
| S3 | MOIT | 95.27 | 69.52 | 94.75 | 94.33 |
| DivideMix | S3 | 95.61 | 46.52 | 95.04 | 94.64 |
| DivideMix | MOIT | 95.75 | 10 | 91.67 | 80.41 |

Table 3: Comparion with other works

## 4.3 SYNTHETIC NOISY DATASETS EVALUATION

In this section, we compared our method to most recent state-of-the-art methods: DivideMix (Li et al., 2020a), LossModelling (Arazo et al., 2019), Coteaching+ (Yu et al., 2019), Mixup (Zhang et al., 2017), F-correction (Patrini et al., 2017), SELFIE (Song et al., 2019), PLC (Zhang et al., 2021), PENCIL (Yi & Wu, 2019), ELR (Liu et al., 2020a), NCT (Chen et al., 2021), MOIT+ (Ortego et al., 2021), NGC (Wu et al., 2021), RRL (Li et al., 2020b), FaMUS (Xu et al., 2021), GJS (Ghosh & Lan, 2021), PDLC (Liu et al., 2020b). We show, that the proposed method achieves consistent improvements in all datasets and at all noise types and ratios.

| Dataset | CIFAR10 | | | | | CIFAR100 | | | |
|---|---|---|---|---|---|---|---|---|---|
| Noise type | Symmetric | | | | Assymetric | Symmetric | | | |
| Noise ratio | 20% | 50% | 80% | 90% | 40% | 20% | 50% | 80% | 90% |
| Cross-Entropy | 86.8 | 79.4 | 62.9 | 42.7 | 85.0 | 62.0 | 46.7 | 19.9 | 10.1 |
| Co-teaching+ | 89.5 | 85.7 | 67.4 | 47.9 | - | 65.6 | 51.8 | 27.9 | 13.7 |
| F-correction | 86.8 | 79.8 | 63.3 | 42.9 | 87.2 | 61.5 | 46.6 | 19.9 | 10.2 |
| Mixup | 95.6 | 87.1 | 71.6 | 52.2 | - | 67.8 | 57.3 | 30.8 | 14.6 |
| PENCIL | 92.4 | 89.1 | 77.5 | 58.9 | 88.5 | 69.4 | 57.5 | 31.1 | 15.3 |
| LossModelling | 94.0 | 92.0 | 86.8 | 69.1 | 87.4 | 73.9 | 66.1 | 48.2 | 24.3 |
| DivideMix | 96.1 | 94.6 | 93.2 | 76.0 | 93.4 | 77.3 | 74.6 | 60.2 | 31.5 |
| ELR | 95.8 | 94.8 | 93.3 | 78.7 | 93.0 | 77.6 | 73.6 | 60.8 | 33.4 |
| AugDesc | 96.3 | 95.4 | 93.8 | 91.9 | 94.6 | 79.5 | **77.2** | 66.4 | 41.2 |
| C2D | 96.4 | 95.3 | 94.4 | 93.6 | 93.5 | 78.7 | 76.4 | 67.8 | **58.7** |
| RRL | 95.8 | 94.3 | 92.4 | 75.0 | 91.9 | 79.1 | 74.8 | 57.7 | 29.3 |
| NGC | 95.9 | 94.5 | 91.6 | 80.5 | 90.6 | 79.3 | 75.9 | 62.7 | 29.8 |
| Ours(S3) | **96.6** | **96.3** | **95.7** | **94.9** | **96.0** | **79.7** | **77.2** | **70.4** | 56.8 |

Table 4: Evaluation of S3 with fixed hyperparameters on CIFAR-10 and CIFAR-100 with closed-set noise. Results of other models are copied from (Li et al., 2020a; Wu et al., 2021).

**Evaluation with controlled closed-set noise** In this section we compare S3 to the most competitive recent works. Table 4 shows results on CIFAR10 and CIFAR100 – we note for S3 this is without the use of co-training such as in C2D (Zheltonozhskii et al., 2021) and without per-dataset finetuning of the hyperparameters as is done in several other methods. Here we report the best results of other methods and the results of S3 with fixed hyperparameters. It is clear that our methods outperforms them, not only in the case of symmetric noise, but also that it works very well also at the more realistic asymmetric synthetic noise.

**Evaluation with combined open-set noise and closed-set noise** Table 5 shows the performance of our method in a more complex combined noise scenario. The closed-set noise are generated as symmetric noise while the open-set noise are random samples from CIFAR100. Noise ratio denotes the total noise ratio while the open ratio denotes the proportion of open-set noise. Previous methods that are specially designed for open-set noise degrade rapidly when the open-set noise ratio

| Method | Noise ratio | | 0.3 | | 0.6 |
|---|---|---|---|---|---|
| | Open ratio | 0.5 | 1 | 0.5 | 1 |
| ILON | Best | 87.4 | 90.4 | 80.5 | 83.4 |
| | Last | 80.0 | 87.4 | 55.2 | 78.0 |
| RoG | Best | 89.8 | 91.4 | 84.1 | 88.2 |
| | Last | 85.9 | 89.8 | 66.3 | 82.1 |
| DivideMix | Best | 91.5 | 89.3 | 91.8 | 89.0 |
| | Last | 90.9 | 88.7 | 91.5 | 88.7 |
| EDM | Best | 94.5 | 92.9 | 93.4 | 90.6 |
| | Last | 94.0 | 91.9 | 92.8 | 89.4 |
| Ours(S3) | Best | **96.34** | **96.05** | **94.97** | **94.01** |
| | Last | **96.13** | **95.95** | **94.81** | **93.54** |

Table 5: Evaluation on CIFAR10 with combined noise. Results of other methods are copied from EDM (Sachdeva et al., 2021).

| Method | | WebVision | | ILSVRC2012 | |
|---|---|---|---|---|---|
| | | Top1 | Top5 | Top1 | Top5 |
| InceptionResNetV2 | DivideMix | 77.32 | 91.64 | 75.20 | 90.84 |
| | ELR | 76.26 | 91.26 | 68.71 | 87.84 |
| | ELR+ | 77.78 | 91.68 | 70.29 | 89.76 |
| | NGC | 79.16 | 91.84 | 74.44 | 91.04 |
| | FaMUS | 79.40 | **92.80** | **77.00** | **92.76** |
| | RRL | 76.3 | 91.5 | 73.3 | 91.2 |
| ResNet50 | GJS | 77.99 | 90.62 | 74.33 | 90.33 |
| ResNet18 | DivideMix | 76.08 | / | / | / |
| | ELR | 73.00 | / | / | / |
| | MOIT+ | 78.76 | / | / | / |
| | Ours(S3) | **80.12** | **92.80** | 74.84 | 91.26 |

Table 6: Testing accuracy on Webvision. Results of other methods are copied from MOIT (Ortego et al., 2021) and NGC (Wu et al., 2021).

is decreased from 1 to 0.5 (Lee et al., 2019; Wang et al., 2018). The performance of method without considering open-set noise like DivideMix (Li et al., 2020a) will decrease when the open-set noise ratio is increased. EDM (Sachdeva et al., 2021) modified the method of DivideMix to deal with combined noise, however report results that are considerably lower than ours.

## 4.4 REAL-WORLD NOISY DATASETS EVALUATION

Finally, in Table 6, Table 7 and Table 8 we show results on the WebVision, Clothing1M and ANIMAL-10N datasets, respectively. To summarize, our method achieves better or comparable performance in relation to the current state-of-the-art in both large-scale web-crawled dataset and small-scale human annotated noisy dataset.

| CE | F-correction | ELR | C2D | FaMUS | RRL | DivideMix* | ELR+* | AugDesc* | Ours(S3) |
|---|---|---|---|---|---|---|---|---|---|
| 69.21 | 69.84 | 72.87 | 74.30 | 74.40 | 74.84 | 74.76 | 74.81 | **75.11** | **74.91** |

Table 7: Testing accuracy on Clothing1M. Results of other methods are from RRL(Li et al., 2020b) and AugDesc (Nishi et al., 2021). Please note methods with * utilized model cotraining or ensembling.

| Cross-Entropy | SELFIE | PLC | NCT | Ours(S3) |
|---|---|---|---|---|
| $79.4 \pm 0.1$ | $81.8 \pm 0.1$ | $83.4 \pm 0.4$ | $84.1 \pm 0.1$ | $\mathbf{88.5 \pm 0.1}$ |

Table 8: Testing accuracy on ANIMAl-10N. Results of other methods are from NCT (Chen et al., 2021).

## 5 CONCLUSIONS

In this paper we propose a method for learning with noisy labels, that relies on a sample selection mechanism, a relabeling mechanism and a training strategy with multi-objective losses that enable us to learn robust features from both noisy and clean samples, and a classifier from only clean or robustly relabeled ones. The proposed method is a simple framework, does not utilize complicated mechanisms such as co-training to deal with self-confirmation bias, and is shown with extensive experiments and ablation studies to be robust to the values of its few hyper-parameters, and to consistently and by large surpass the state-of-the art in both open-set and close-set noise.

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

## A  DATASET DETAILS

CIFAR10 and CIFAR100 both consist of 50K images. Following the standard practice, for CIFAR-10 and CIFAR-100, we experimented with two types of artificial noise: *symmetric noise* by randomly replacing labels of all samples using a uniform distribution; and *asymmetric noise* by randomly exchanging labels of visually similar categories, such as Horse $\leftrightarrow$ Deer and Dog $\leftrightarrow$ Cat.

WebVision (Li et al., 2017) is a large-scale dataset of 1000 classes of images crawled from the Web. Following previous work (Jiang et al., 2018; Li et al., 2020a; Ortego et al., 2021), we compare baseline methods on the top 50 classes from Google images Subset of WebVision. The noise ratio that is estimated to be around 20%. ANIMAL-10N (Song et al., 2019) is a smaller and recently proposed real-world dataset consists of 10 classes of animals, that are manually labeled with an error rate that is estimated to be approximately 8%. ANIMAL-10N has similar size characteristics to the CIFAR datasets, with 50000 train images and 10000 test images.

## B  EXPERIMENT DETAILS

We used a PresActResNet-18 (He et al., 2016) as the backbone for all CIFAR10/100 experiments following previous works. Unlike previous methods that use specific warmup settings for CIFAR10/CIFAR100, we train the model from scratch with a linear raising $\theta_s$ from 0 to 1 in 20 epochs. $\theta_r$ is fixed as 0.8 and the prediction track length $L$ is set to 10 for all CIFAR experiments except in the corresponding ablation part. We train all modules with the same SGD optimizer for 300 epochs with a momentum of 0.9 and a weight decay of 5e-4. The initial learning rate is 0.02 and is controlled by a cosine annealing scheduler by Pytorch. The batchsize is fixed as 128. We set $\alpha = 4$ for all noise settings in Mixup training.

For WebVision, we used a standard ResNet18 following MOIT (Ortego et al., 2021) due to the hardware limitation. We train the network with SGD optimizer for 150 epochs with momentum

of 0.9 and a weight decay of 1e-4. The initial learning rate is 0.02 and is controlled by a cosine annealing scheduler. The batchsize is fixed as 64. For Clothing1M, we used ResNet50 following DivideMix (Li et al., 2020a) with ImageNet pretrained weights. We train the network with SGD optimizer for 80 epochs with momentum of 0.9 and weight decay of 1e-3. The initial learning rate is 0.02 and and is reduced to 0.002 after 40 epochs. For ANIMAL-10N, we used VGG-19 (Simonyan & Zisserman, 2014) with batch-normalization following (Song et al., 2019). We train the network with SGD optimizer for 100 epochs with momentum of 0.9 and weight decay of 5e-4. The initial learning rate is 0.02 and and is also controlled by a cosine annealing scheduler. The batchsize is fixed as 128. For all datasets, we train the model from scratch with $\theta_s = 1$, while $\theta_r$ is fixed as 0.9 and the prediction track length $L$ is set to 1. We set $\alpha = 1$ for Mixup. We report averages of at least two runs on a single Nvidia RTX 3090 GPU card.

## C EFFECT OF AUGMENTATIONS

In Table 9, we show the effect of different types of augmentations at the different stages of our method. Our results are consistent with the findings of AugDesc (Nishi et al., 2021) that one should use weaker augmentations for sample selection and relabeling, and strong augmentations for training. We can also see that the higher the noise ratio, the more prominent the differences are with different augmentation strategies.

| Dataset | 50% sym CIFAR10 | | | | | | 90% sym CIFAR10 | | | | | |
|---|---|---|---|---|---|---|---|---|---|---|---|---|
| SSR | N | | W | | S | | N | | W | | S | |
| SST | W | S | W | S | W | S | W | S | W | S | W | S |
| ACC (%) | 96.18 | 96.45 | 96.41 | **96.48** | 96.14 | 96.22 | 93.46 | **95.13** | 93.70 | 94.92 | 90.92 | 93.85 |
| AugDesc (Nishi et al., 2021) | 95.6 | | | | | | 91.9 | | | | | |

Table 9: Classification accuracy for different augmentations. N denote 'none' augmentation, W denote 'weak' augmentation, S denote 'strong' augmentation; SSR denote sample selection&relabelling, SST denote supervised self-supervised training.

## D DIFFERENT DISTANCE METRICS IN SELF-CONSISTENCY

In S3, we applied negative cosine similarity as distance metric for self-consistency head. Here we also experimented with L2 distance. The results are shown in Table 10, where it can be seen that there are small differences, but that the cosine similarity is in general better.

| Training | 50% sym | 90%sym | 40% asym | 60% all(50% open-set) |
|---|---|---|---|---|
| S3(negative cosine similarity) | **96.30** | **95.20** | 96.0 | **94.81** |
| S3(L2 distance) | 96.04 | 94.7 | **96.13** | 94.26 |

Table 10: S3 with different distance metric for self-consistency loss

