# OpenReview forum: "S3: Supervised Self-supervised Learning under Label Noise"
_ICLR.cc/2022/Conference — ICLR 2022 Submitted_

### Official Review · Reviewer_pgYs · 2021-10-20

**Correctness:** 3
**Technical Novelty And Significance:** 1
**Empirical Novelty And Significance:** 1
**Recommendation:** 5
**Confidence:** 4

**Main Review:**

The novelty of this paper is ad-hoc, which stacks the benefits of sample selection and contrastive learning. Considering several previous works have explored them [1, 2, 3] and the proposed method in this paper has minor difference with previous works, it is below the bar of ICLR for the area of learning with noisy labels.

[1] Label Distribution for Learning with Noisy Labels. IJCAI 2020
[2] Contrastive Learning Improves Model Robustness Under Label Noise. CVPRW 2021
[3] Learning from Noisy Data with Robust Representation Learning. ICCV 2021

**Summary Of The Paper:**

This paper focuses on learning with noisy labels problems, which constructs an iterative learning framework to refine labeling set and train the model parameters.  It demonstrates its advantages by comparing with current baselines on a range of datasets.

**Summary Of The Review:**

This paper focuses on learning with noisy labels problems, which constructs an iterative learning framework to refine labeling set and train the model parameters.  It demonstrates its advantages by comparing with current baselines on a range of datasets. However, the novelty of this paper is ad-hoc, which stacks the benefits of sample selection and contrastive learning. Considering several previous works have explored them [1, 2, 3] and the proposed method in this paper has minor difference with previous works, it is below the bar of ICLR for the area of learning with noisy labels. Another problem is k-NN search for neighborhood voting is computational-expensive especially for large-scale datasets.

[1] Label Distribution for Learning with Noisy Labels. IJCAI 2020
[2] Contrastive Learning Improves Model Robustness Under Label Noise. CVPRW 2021
[3] Learning from Noisy Data with Robust Representation Learning. ICCV 2021

---

> ### Author Response · Authors · 2021-11-18
> **Response to Reviewer pgYs(Part 1)**
>
> > **Q3.1** ‘’The novelty of this paper is ad-hoc, which stacks the benefits of sample selection and contrastive learning. Considering several previous works have explored them [1, 2, 3] and the proposed method in this paper has minor difference with previous works, it is below the bar of ICLR for the area of learning with noisy labels.''
>
> We respectfully disagree. To train efficiently under different types of label noise, one needs a selection/relabelling mechanism that selects/relabels samples (sec 3.3), and a training framework that utilizes both clean and noisy samples (sec. 3.4). All SOTA works in the field have implicitly or explicitly those components but sometimes mix those two stages.
>
>    The main novelty of the proposed method is that it totally decouples these stages and uses a) off-the-shelf training methods (section 3.4) without bells and whistles (like co-training and ensembles), and b) a selection and relabelling strategy that can be expressed in just 3 equations (section 3.3) and with 3 hyperparameters (to which it is quite robust as we show in the ablation studies).
>
>    Please see our response to Reviewer cqfk (Q2.1 and Q2.3) including the ablation studies that demonstrate how much each part contributes to the performance.
>
>
>    We would also like to clarify that our method does not use contrastive learning. Moreover, the works that the reviewer mentions are interesting but have important shortcomings:
>
>    [1] estimates the reliability of the label of a sample based on the labels of its neighbors in a trusted/clean set. However, it is not realistic to assume that there exists a (trusted) subset with clean labels. [1] also adopts the semi-supervised training of DivideMix [4], where samples in the untrusted/noisy set are implicitly relabelled with the label that the model/classifier assigns to them -- this is detrimental as we explain (and experimentally show) in the response in Q2.1 and Q2.3.
>
>    [2] is an interesting work that is very similar to C2D [5]: it uses a pre-training using self-supervision so as to arrive to good representations, and then applies off-the-shelf methods for training with noisy labels. The results support our conclusions about the importance of training using a loss that ignores the labels. In their case this is just used in a pre-training stage, while in ours is incorporated in an end-to-end methodology that deals with the heart of the problem. For example, the method that [2] uses after the pre-training, that is (MWNet), deals with noisy samples by reweighing them and therefore it cannot handle open-set-noise or relabel samples so as to learn from them as well.
>
>    [3] This is an interesting work, however, in the heart of it is a relabelling mechanism based on the labels of the nearest neighborhoods and a contrastive learning based on class prototypes that has the shortcoming of methods that rely on relabeling (implicitly or not) of all the samples. Among the weaknesses is that relabeling all samples is problematic in the case of open-set noise. The method is also rather complex with mix-up training using class prototypes and co-training.
>
>    For these reasons all the above works perform worse, sometime by a large margin in comparison to our method. This can be seen in Table Q3.1  below. We will include results and discussion in the paper. Please note here we don't include results from [1] as they are out of SOTA and require extra clean dataset.
>
> | Method | 20% sym (CIFAR10) | 50% sym (CIFAR10) | 60% sym (CIFAR10) | 80% sym (CIFAR10) | 90% sym (CIFAR10) | 40% asym (CIFAR10) | 20% sym (CIFAR100) | 50% sym (CIFAR100) | 60% sym (CIFAR100) | 80% sym (CIFAR100) | 90% sym (CIFAR100) |
> |--------|-------------------|-------------------|-------------------|-------------------|-------------------|--------------------|--------------------|--------------------|--------------------|--------------------|--------------------|
> | RRL    | 95.8              | 94.3              | /                 | 92.4              | 75.0              | 91.9               | 79.1               | 74.8               | /                  | 57.7               | 29.3               |
> | GJS    | 95.3              | /                 | 91.64             | /                 | /                 | 89.65              | /                  | /                  | 73.6               | /                  | /                  |
> | S3            | **96.6** | **96.3** | /                 | **95.7** | **94.9** | **96.0** | **79.7** | **77.2** | /                  | **70.4** | **56.8** |
>
> *Table Q3.1: Comparisons with RRL [3], and GJS [2]*

---

> ### Author Response · Authors · 2021-11-18
> **Response to Reviewer pgYs(Part 2)**
>
> > **Q3.2** Another problem is k-NN search for neighborhood voting is computational-expensive especially for large-scale datasets.
>
> Thanks for your comment. Please see our response to Reviewer cqfk (Q2.2)  including the training time analysis especially the kNN operation in all datasets. It is clear that the kNN-based sample selection takes very little time compared to the Training stage.
>
> *[1] Liu, Yun-Peng, et al. "Label Distribution for Learning with Noisy Labels." _IJCAI_. 2020.*
>
> *[2] Ghosh, Aritra, and Andrew Lan. "Contrastive Learning Improves Model Robustness Under Label Noise." _Proceedings of the IEEE/CVF Conference on Computer Vision and Pattern Recognition_. 2021.*
>
> *[3]  Li, Junnan, Caiming Xiong, and Steven CH Hoi. "Learning From Noisy Data With Robust Representation Learning." _Proceedings of the IEEE/CVF International Conference on Computer Vision_. 2021.*
>
> *[4] Li, Junnan, Richard Socher, and Steven CH Hoi. "DivideMix: Learning with Noisy Labels as Semi-supervised Learning." _International Conference on Learning Representations_. 2019.*
>
> *[5] Zheltonozhskii, Evgenii, et al. "Contrast to Divide: Self-Supervised Pre-Training for Learning with Noisy Labels." _arXiv preprint arXiv:2103.13646_ (2021).*

---

### Official Review · Reviewer_cqfk · 2021-10-20

**Correctness:** 3
**Technical Novelty And Significance:** 2
**Empirical Novelty And Significance:** 2
**Recommendation:** 5
**Confidence:** 4

**Main Review:**

Strength:

1.  The paper is easy to follow. The proposed "S3"  framework achieves very good performance on both close-set label noise and open-set label noise.

2. The ablation studies are well designed. Each part of S3 along with the hyper-parameters are analyzed.

Concerns:

1. I do not think "S3" is a simple framework or simple method. It contains two stages with multiple techniques such as KNN voting, relabelling, sample selection, mixup and consistency loss, which make it hard to understand which part exactly contributes most to the performance. Further, I am worried that the techniques such as KNN voting may increase the training time. Authors are encouraged to calculate the training time of each module in S3.

2. KNN voting-based relabelling, mixup and consistency loss have already been explored in the literature of learning with noisy labels [A1, A2, A3].  I do admire the performance gain brought by S3 in the experiments. However, without deeper analyses, I think the technical novelty is a little limited since it seems that S3 simply combines existing approaches with little modifications. To improve the paper, authors are encouraged to elaborate the main difference among S3 and other approaches in [A1, A2, A3] and state why the combination or the modification is important.

3.  From Figure 2 (a), the network still has good performance when $\theta_{r}$ approximates 1. Does it mean that relabelling is not very necessary? I think it will be much better if authors calculate the F-score to represent the quality of sample selection and relabelling especially when open-set label noise involved.



Reference:

A1: Multi-Objective Interpolation Training for Robustness to Label Noise. CVPR 2021

A2: Unsupervised Label Noise Modeling and Loss Correction. ICML 2019

A3: Consistency Regularization Can Improve Robustness to Label Noise. ICML 2021



**Summary Of The Paper:**

This paper proposes "S3" framework for learning with noisy labels. Specifically, S3 consists of two stages. In the first stage, a relabelling approach and normalized neighboring voting are utilized to guide efficient sample selection; in the second stage, supervised loss (Mixup) and self-consistency loss are used to train networks on selected samples. S3 can be applied to both close-set label noise and open-set label noise and exhibit good performance on several benchmark datasets.

**Summary Of The Review:**

**Pre-rebuttal:** S3 achieves very good performance but the technical novelty is limited. In the current version, it seems that S3 is more like a combination work. Thus I give the initial score of 5. I will consider increasing my score if authors can well clear my concerns.


**Post-rebuttal & Final recommendation:**

I thank the authors for their further reply and respect to other works.

It is relatively a hard decision for me. Experimentally, this paper does achieve promising performance. The authors did a good literature review in the paper
and elaborate the difference among S3 and other methods in the rebuttal. The designed S3 framework only has few hyper-parameters which are shown robust. To this, I appreciate the efforts and contribution.  However, I still think the combination of components of S3 lacks enough novelty since each component is not new in the literature.
For the DivideMix that the authors mention, I admit that DivideMix is also a combination paper. However, the performance of DivideMix
consistently outperforms all the previous methods by a large margin especially when the noise rate is high (at that time). While it seems S3 only has little improvement compared to C2D, which is also a work that uses self-supervised learning. In some cases, C2D is even better. Further, considering the supervised, self-supervised training framework is also explored in other works (Co-learning) and S3 currently does not have theoretical justifications such as why KNN voting-based relabelling, or other component, works on both close-set and open-set label noise, I remain my final score as 5.

---

> ### Author Response · Authors · 2021-11-18
> **Response to Reviewer cqfk(Part 1)**
>
> > **Q2.1**     "I do not think "S3" is a simple framework or simple method. It contains two stages with multiple techniques such as KNN voting, relabelling, sample selection, mixup and consistency loss, which make it hard to understand which part exactly contributes most to the performance."
>
> To train efficiently under different types of label noise, one needs a selection/relabelling mechanism that selects/relabels samples (sec 3.3), and a training framework that utilizes both clean and noisy samples (sec. 3.4). All SOTA works in the field have implicitly or explicitly those components but sometimes mix those two stages.
>
> On the contrary our framework is very simple. It totally decouples these stages and uses a) off-the-shelf training methods (section 3.4) without bells and whistles (like co-training and ensembles), and b) a selection and relabelling strategy that can be expressed in just 3 equations (section 3.3) and with 3 hyperparameters (to which it is quite robust as we show in the ablation studies).
>
> We do appreciate the reviewer's comment and in order to demonstrate how much each part contributes to the performance we perform experiments where substitute a) our Sample/Relabeling method and/or b) the training strategy with components of the methods that the reviewer suggests (A1: MOIT [1] , A2: LossModelling [2] , A3: GJS [3] ). Please see our additional ablations in our response to Q2.3.
>
> We will add the discussion below in the paper to clarify further:
>
> - A) The sample selection and relabeling (section 3.3) are designed so as to construct a clean subset that has as many correctly labelled samples as possible so as to train with a supervised learning cost in Sec.3.3. We chose the sample selection mechanism to be based on k-NN neighbors in order to reduce the self-confirmation bias -- we show that this is better than selecting samples based on the supervised learning loss (like A2 and DivideMix [4] does) in our new experiments (Table Q2.3.1). We also show the effect of relabelling (Fig.1 in original submitted paper) where we report the results with different levels of relabelling.
>
> - B) For learning (section 3.4), we claim that it is essential to have both a supervised learning loss on the clean subset (cross entropy) and an unsupervised learning loss (consistency) that makes no assumption at all (implicit or explicit) about the labels of the noisy subset. We claim that this is better than using the semi-supervised training scheme that papers such as DivideMix , MOIT and their derivatives (e.g. AugDesc [5]) use, in which they implicitly relabel the noisy samples with the soft labels that their classifier/model gives. Such schemes do not perform well under heavy noise or under open-set noise because the soft labels that are used in the training are contaminated by noise and the feature extractor cannot be trained well. The observation that a self-supervised representation learning scheme helps is supported by the results of C2D [6] , where a pre-training with constrastive learning followed by DivideMix gave strong results.
>
> *[1]  Diego Ortego, Eric Arazo, Paul Albert, Noel E O’Connor, and Kevin McGuinness. Multi-objective interpolation training for robustness to label noise. In Proceedings of the IEEE/CVF Conference on Computer Vision and Pattern Recognition, pp. 6606–6615, 2021.*
>
> *[2] Arazo, Eric, et al. "Unsupervised label noise modeling and loss correction." _International Conference on Machine Learning_. PMLR, 2019.*
>
> *[3] Englesson, Erik, and Hossein Azizpour. "Consistency Regularization Can Improve Robustness to Label Noise." _arXiv preprint arXiv:2110.01242_ (2021).*
>
> *[4] Li, Junnan, Richard Socher, and Steven CH Hoi. "DivideMix: Learning with Noisy Labels as Semi-supervised Learning." _International Conference on Learning Representations_. 2019.*
>
> *[5] Nishi, Kento, et al. "Augmentation strategies for learning with noisy labels." _Proceedings of the IEEE/CVF Conference on Computer Vision and Pattern Recognition_. 2021.*
>
> *[6] Zheltonozhskii, Evgenii, et al. "Contrast to Divide: Self-Supervised Pre-Training for Learning with Noisy Labels." _arXiv preprint arXiv:2103.13646_ (2021).*

---

> ### Author Response · Authors · 2021-11-18
> **Response to Reviewer cqfk(Part 2)**
>
> > **Q2.2** ''Further, I am worried that the techniques such as KNN voting may increase the training time. Authors are encouraged to calculate the training time of each module in S3.''
>
> Thank you for the comment. kNN is performed after every epoch similar to other works that perform metric learning or label analysis in local neighborhoods and works such as MOIT. Below, we report the running time of each module on the datasets that we have experimented.
>
> |    Dataset(Trainset size)        | Training | Forward pass(Feature extraction) | kNN sample selection |
> |------------|----------|--------------------------------|----------------------------------|
> | CIFAR(50k)      | 112s     | 9s                             | 1.23s                            |
> | WebVision(~65k)  | 587s     | 109s                           | 1.48s                           |
> | Clothing1M(32k) | 575s     | 57s                            | 0.79s                            |
> *Table Q2.2: Training time analysis*
>
> It is clear that the kNN-based sample selection takes very little time compared to the Training stage. Please note that the k-NN neighborhood computations are even a fraction of the time that is needed for a forward pass (feature extraction, column 2) of the model and that there are efficient methods for k-NN search that we haven't employed.

---

> ### Author Response · Authors · 2021-11-18
> **Response to Reviewer cqfk(Part 3)**
>
> > **Q2.3**  ''KNN voting-based relabelling, mixup and consistency loss have already been explored in the literature of learning with noisy labels [A1, A2, A3]. I do admire the performance gain brought by S3 in the experiments. However, without deeper analyses, I think the technical novelty is a little limited since it seems that S3 simply combines existing approaches with little modifications. To improve the paper, authors are encouraged to elaborate the main difference among S3 and other approaches in [A1, A2, A3] and state why the combination or the modification is important.''
>
> Thank you for recommending three related papers. We were aware of A1: MOIT and A2: LossModelling at submission stage and cited them in our paper and we will cite A3: GJS as well.  The works that the reviewer mentions (A1, A2 and A3) are important and interesting but have different, important shortcomings.
>
> The sample selection and relabelling mechanisms that they adopt are suboptimal in comparison to ours. GJS, does not have any selection or relabelling mechanism. LossModelling relabels all samples that it deems noisy, with one of the labels in the training set. In MOIT, the semi-supervised learning scheme unlabelled samples are implicitly assumed to have the soft labels that the model assigns to them. While these are important works, it is essential in our view to not try to relabel or make assumptions about the labels of all samples in the dataset. In addition, A3 imposes consistency in the label space -- this is not optimal in the presence of samples whose real label is not one of those in the training set, that is, it does not deal with open set noise.
>
> As a result, these methods have by large lower performance than our method, especially in the case of large amounts of noise. This is clear in Table Q2.3.1 where for 90\% symmetric noise, our method outperforms A1 by 18.9\% and A2 by 25.8\%. (Please note, in (A1) MOIT, the noise ratio is generated in a different setting: in most of works 90\% symmetric noise corresponds to 90\% $\times$ 0.9 = 81\% noise ratio in MOIT. Here we compare MOIT results under 40\% and 80\% symmetric noise with S3 under 50\% (~45\% in MOIT) and 90\% to have a fair comparison).
>
> | Method        | 20% sym (CIFAR10) | 50% sym (CIFAR10) | 60% sym (CIFAR10) | 80% sym (CIFAR10) | 90% sym (CIFAR10) | 40% asym (CIFAR10) | 20% sym (CIFAR100) | 50% sym (CIFAR100) | 60% sym (CIFAR100) | 80% sym (CIFAR100) | 90% sym (CIFAR100) |
> |---------------|-------------------|-------------------|-------------------|-------------------|-------------------|--------------------|--------------------|--------------------|--------------------|--------------------|--------------------|
> | MOIT          | /                 | 94.1              | /                 | /                 | 75.8              | 93.27              | /                  | 75.9               | /                  | /                  | 51.4               |
> | LossModelling | 94.0              | 92.0              | /                 | 86.8              | 69.1              | 87.4               | 73.9               | 66.1               | /                  | 48.2               | 24.3               |
> | GJS           | 95.3              | /                 | 91.6             | /                 | /                 | 89.65              | 78.1                  | /                  | 70.2               | /                  | /                  |
> | S3            | **96.6** | **96.3** | /                 | **95.7** | **94.9** | **96.0** | **79.7** | **77.2** | /                  | **70.4** | **56.8** |
>
> *Table Q2.3.1  : Comparisons with (A1) MOIT , LossModelling(A2) and GJS(A3)*
>
> #### Additional ablations
>
> To illustrate the benefits of the choices that we make we report results in Table Q2.3.2 (In response Part 4).
>  - a) S3 sample selection and MOIT (A1) training with (implicit) relabeling and semi-supervised training. The deterioration in the performance at high noise ratios is clear.
>  - b) A loss-based selection mechanism and our Supervised Self-supervised (S3) training scheme. The results show that the k-NN based selection is better in avoiding the self-confirmation bias (compare with S3), but also show how much better our relabeling and training is compared to the (implicit) relabeling and semi-supervised training that A2 adopts (compare with A2 in table ).
> -  c) Loss-based selection (A2) and MOIT (A1) training with (implicit) relabeling and semi-supervised training. The comparison to a) shows again the superiority of k-NN based selection.

---

> > ### Comment · Reviewer_cqfk · 2021-11-24
> > **Still have concerns about the novelty**
> >
> > Thanks for such a detailed response which addresses part of my concerns. I am convinced that the combination of techniques such as KNN voting-based relabelling, mixup, and consistency loss does improve the model robustness to a very high level in both close-set and open-set label noise. The revised submission also improves the paper. However, my main concern about novelty still remains.
> >
> > In my initial review, I listed some papers relevant to your method "S3". I also noticed some papers pointed by other reviewers. The author's response has well explained the differences among S3 and other approaches and I am convinced by the explanation. However, I actually expect deeper analysis such as a theoretical understanding on the combination of some components in S3. Note theoretical understanding is not a must to me. But it is hard for me to find a single component in S3 that is novel enough or leads to very interesting observations (for ICLR level from my understanding). Thus I hope authors can provide more interesting and important (theoretical) justification for their method.
> >
> > Since KNN voting-based relabeling is the beginning of S3, I think authors can follow the theoretical framework from [R1] to perform further interesting analyses. For example, how KNN behaves for open-set label noise or if there exists some optimal hyper-parameters such as $\theta_{r}$ that can be calculated or expressed. Of course, this is only my advice. But I think some theoretical justification can make S3 more appealing.
> >
> >
> > Additional comments:
> >
> > - "F-score" in my initial review is a criterion to test the quality of sample selection. For example, Figure 3 in [R2], Figure 4 in [R3].
> >
> > [R1] Deep KNN for Noisy Labels. ICML 2020
> >
> > [R2] Learning with instance-dependent label noise: A sample sieve approach. ICLR 2021.
> >
> > [R3] FINE Samples for Learning with Noisy Labels. NeurlPS 2021.

---

> > > ### Author Response · Authors · 2021-11-26
> > > **Further reply**
> > >
> > > Thank you for your reply – we do appreciate a lot the engagement and constructive suggestions of the reviewer.
> > >
> > > We would like to point out that the theoretical analysis in [R1, R2, R3] is very interesting, however, it is not very clear to us what is the practical significance of the bounds that are derived. Those are derived on single components which, however, give performances that are far from the SOTA. We are not aware of SOTA methods, or close to SOTA methods that give bounds. Taking as an example [R3] which performs the best among [R1,R2,R3], we note that the confidence measure that they propose needs to be embedded in a DivideMix [1] sample selection and training scheme – even so, their results are close to the original DivideMix and by far lower than ours on heavy noise. For example their accuracy in CIFAR-100 @80% and @90% symmetric noise at 60.1% and 31.2% and ours at 70.4% and 56.8%, respectively.
> > >
> > > Regarding the novelty of our scheme, we would like to point the reviewer to our response to comment pgYs (Q3.1) and our ablation studies in Table 3 (revised paper). It is clear that the success of our method does not rely on a single component but in the specific combination of the choices we make in the two stages – for those combinations it is difficult to derive theoretical bounds.
> > >
> > > However, we believe there are a few important observations one could make:
> > >
> > > - First, that training in the supervised self supervised manner that we propose is better than training in the semi-supervised manner that most methods in the literature (e.g. MOIT [2], DivideMix, LossModelling [3], SELF [4] .etc) use. We propose to use the supervised, self-supervised framework for training – we are the first to do so in this context, and the importance of doing so is clear from the experiments we believe. There is a clear motivation: with semi-supervision noisy samples are implicitly relabeled and used albeit with small contributions – when the noise is heavy, their collective influence is large. We do not have theoretical bounds on their influence, but it is clear from figure 4(b) and that when the size of the set with correct labels is small (e.g. red line, noise @90%) the performance of semi-supervised MOIT method (Table 3, 2nd row) is affected the most. As we said, we are the first to do this type of training in this context and that the training is with off-the-shelf modules without bells and whistles – For example DivideMix used cotraining and a regularisation term to avoid the model collapsing to predicting only one class.
> > >
> > > - Second, that the kNN based sample selection scheme that we propose is better than the loss-based ones proposed in (among others) DivideMix. This is clearly shown in the results in Table 3. It is also clear that it performs better than the new methods that the reviewer mentions, i.e., [R2], [R3], if one compares the F-score figures that they report (Fig.3 in [R2], Fig.4 in [R3] and Fig.4 in our revised paper). Our sample selection/relabeling is extremely simple using a kNN and just three equations. By contrast the selections that are obtained with [R3] for example, need eigenvalue decomposition of the Gram matrix. We think that the simplicity of our method is an advantage.
> > >
> > > Some additional points we would like to make:
> > >
> > > - An appropriate prior for modelling noise generation is a prerequisite for theoretical analysis, and works that do so, make the assumption of symmetric or asymmetric label transition table – this may not be realistic in open-set noise or in real datasets.
> > >
> > > - Compared to a loss-based selection method that requires a classifier head that is reasonably well-trained via a warmup stage with all samples, our relabeling/selection starts from scratch. Also, loss-based methods need to set the number of warmup stage epochs so as to not overfit to noise and adjust to the difficulty of the dataset -- this is not trivial. kNN-based selection only require local smoothness in the features space and works better in high noise ratio.
> > >
> > > References:
> > >
> > > _[1] Li, Junnan, Richard Socher, and Steven CH Hoi. “DivideMix: Learning with Noisy Labels as Semi-supervised Learning.”  _International Conference on Learning Representations_. 2019._
> > >
> > > _[2] Diego Ortego, Eric Arazo, Paul Albert, Noel E O’Connor, and Kevin McGuinness. Multi-objective interpolation training for robustness to label noise. In Proceedings of the IEEE/CVF Conference on Computer Vision and Pattern Recognition, pp. 6606–6615, 2021._
> > >
> > > _[3] Arazo, Eric, et al. “Unsupervised label noise modeling and loss correction.”  _International Conference on Machine Learning_. PMLR, 2019._
> > >
> > > _[4] Nguyen, Duc Tam, et al. “Self: Learning to filter noisy labels with self-ensembling.”  _arXiv preprint arXiv:1910.01842_  (2019)._

---

> > > > ### Comment · Reviewer_cqfk · 2021-11-27
> > > > **Further comments**
> > > >
> > > > To be clear, I do not request a theoretical analysis on S3 currently. I know it is hard to build a theory in a short time. However, if authors can provide such one (single component or combination of components). I would be very happy to check and improve the score. As I said in my last response, theoretical analysis is not a must to me. I mention the theoretical part just because I feel the combination of the components in S3 lacks enough novelty and I hope some theoretical justification can further improve the paper. I thank the authors for spending time to respond to my further concerns. But I have some disagreements to your reply.
> > > >
> > > >
> > > > - I do not think that the bounds (of a single component) is not practical just because the performance (of a single component) is not SOTA or far away from SOTA. It is well known that one can add many tricks to improve the performance of a single component such as semi-supervised training, mixup, early stopping, etc, while the bounds and theoretical analyses can facilitate a better understanding on the limitation and optimal condition of the component.
> > > >
> > > > - I also do not think all SOTA methods lack theoretical analysis. For example, [R2] achieved SOTA performance on instance-dependent label noise (at that time). It also has two stages (sample selection + semi-supervised consistency training). But the component of sample selection is well theoretically analyzed including the hyper-parameters and the bounds.
> > > >
> > > > - Authors state that S3 can achieve remarkable performance on the extreme noise level. However, C2D also achieves very good performance on extreme noise levels (even on 95% noise rate in their paper). Besides, it is arguable that S3 is the first paper that uses the supervised, self-supervised framework for training. Co-learning (https://arxiv.org/pdf/2108.04063v1.pdf) also deploys such a framework.
> > > >
> > > > Additional Comments
> > > >
> > > > - It is true that for some theoretical analyses, we need to assume some appropriate priors. For example, we assume the label noise is symmetric or asymmetric. But such prior is often used to prove a consistent classifier. If we try to analyze the module of sample selection, such priors on label noise generation may not be needed [R1, R3] or can be very realistic (instance-dependent, [R2]).

---

> > > > > ### Author Response · Authors · 2021-11-29
> > > > > **Further reply (Part 1)**
> > > > >
> > > > > > **Concerns about novelty and motivation**  "I mention the theoretical part just because I feel the combination of the components in S3 lacks enough novelty and I hope some theoretical justification can further improve the paper"
> > > > >
> > > > > We appreciate the reviewer's viewpoint, but would like to re-iterate the main motivation and novelty of our approach:
> > > > >
> > > > > - Motivation: To train efficiently under different types of label noise, one needs a selection/relabelling mechanism that selects/relabels samples (sec 3.3), and a training framework that utilizes both clean and noisy samples (sec. 3.4). All SOTA works in the field have implicitly or explicitly those components but sometimes mix those two stages and make suboptimal choices.
> > > > >
> > > > > - Novelty: The main novelty of the proposed method is that it totally decouples these stages and uses a) off-the-shelf training methods (section 3.4) without bells and whistles (like co-training and ensembles), and b) a selection and relabelling strategy that can be expressed in just 3 equations (section 3.3) and with 3 hyperparameters (to which it is quite robust as we show in the ablation studies).
> > > > >
> > > > > Although some of the techniques used in our method can be found in other contemporary or much older papers as well, the way that those ideas are brought together is novel and the differences from the works that use those ideas, are very well motivated, clear, and prove to be important in practice. As commented by reviewer bqh4 ('While the ideas of neighborhood-based sample selection, sample relabeling, and consistency regularization are well-known, the particular implementation is novel and interesting.'). We end up with a single, simple framework, with well motivated choices, with very few hyperparameters that achieves very good results -- we believe that this is significant enough for a top-level conference.
> > > > >
> > > > > Finally, we would like to bring up the  well established DivideMix [1], which appeared frequently in the previous discussion. DivideMix combined Loss-based sample selection [2] with semi-supervised learning, (more specifically using MixMatch [3]) which are well-known methods. Nevertheless  DivideMix put these components together in a novel way and, since then, has motivated several works in exploring learning under label noise. We expect S3 will motivate others on how to deal with openset noise, sample selection choices and use of self-supervision.
> > > > >
> > > > > > **Regarding the relations with works on self-supervision (C2D and Co-learning)** " Authors state that S3 can achieve remarkable performance on the extreme noise level. However, C2D also achieves very good performance on extreme noise levels (even on 95% noise rate in their paper). Besides, it is arguable that S3 is the first paper that uses the supervised, self-supervised framework for training. Co-learning ([https://arxiv.org/pdf/2108.04063v1.pdf](https://arxiv.org/pdf/2108.04063v1.pdf)) also deploys such a framework."
> > > > >
> > > > > C2D [4] is an Arxiv paper, that does indeed have a very strong performance, and is an important work in showing the value of self-supervision. However, they use self-supervision only as pre-training. After the pre-training of the self-supervision, they apply a DivideMix which applies implicit relabeling of noisy samples during training based on a loss-based selection. Theoretically this is problematic in the case of out-of-set noise but also at heavy-noise where the classification head gets wrong supervision signal from noisy-labeled samples (albeit reduced) especially in early iterations.
> > > > >
> > > > > The co-learning paper (https://arxiv.org/pdf/2108.04063v1.pdf) that the reviewer mentions is interesting, and indeed they do supervised, self-supervised learning, however, and crucially, they do not have a sample relabeling/selection stage. Their performance is by far below the SOTA as a result. We didn't know it at the time of the submission, but we will cite it. There are also a few more drawbacks in our view -- for example the self-supervision on structural loss is on the classification outputs and not on the feature space.

---

> > > > > > ### Author Response · Authors · 2021-11-29
> > > > > > **Further reply (Part 2)**
> > > > > >
> > > > > > > **Regarding the value of theoretical analysis** " I do not think that the bounds (of a single component) is not practical just because the performance (of a single component) is not SOTA or far away from SOTA. It is well known that one can add many tricks to improve the performance of a single component such as semi-supervised training, mixup, early stopping, etc, while the bounds and theoretical analyses can facilitate a better understanding on the limitation and optimal condition of the component. I also do not think all SOTA methods lack theoretical analysis. For example, [R2] achieved SOTA performance on instance-dependent label noise (at that time). It also has two stages (sample selection + semi-supervised consistency training). But the component of sample selection is well theoretically analyzed including the hyper-parameters and the bounds.”
> > > > > >
> > > > > > We accept the points that the reviewer makes, and we see that we could have phrased our response differently: Indeed meaningful theoretical analysis does have value and we respect the quality of the analysis, and the work in all of the papers that the reviewer mentioned. Specifically with respect to [R2], the novelty is indeed on a proposed regularisation term, and on a selection mechanism with bounds derived when minimizing the Empirical Risk, and on the selection mechanism -- however, SOTA results are when training in a semi-supervised manner with a consistency term. Similarly, the bounds that [R3] derives are on the selection mechanism, however, the same selection mechanism in different training schemes gives very different performance. What we are pointing out is that theoretical analysis in such mixed training scheme is hard.
> > > > > >
> > > > > > We regret that we cannot give proofs of bounds in a short time, and hope to do some meaningful theoretical analysis in our subsequent work -- we believe that the interplay between the selection mechanism and the supervised, self-supervised training is an interesting direction.
> > > > > >
> > > > > > *[1] Li, Junnan, Richard Socher, and Steven CH Hoi. “DivideMix: Learning with Noisy Labels as Semi-supervised Learning.”  _International Conference on Learning Representations_. 2019.*
> > > > > >
> > > > > > _[2] Arazo, Eric, et al. “Unsupervised label noise modeling and loss correction.”  _International Conference on Machine Learning_. PMLR, 2019._
> > > > > >
> > > > > > *[3] Berthelot, David, et al. "MixMatch: A Holistic Approach to Semi-Supervised Learning." _Advances in Neural Information Processing Systems_ 32 (2019).*
> > > > > >
> > > > > > *[4] Zheltonozhskii, Evgenii, et al. "Contrast to Divide: Self-Supervised Pre-Training for Learning with Noisy Labels."  _arXiv preprint arXiv:2103.13646_  (2021).*

---

> ### Author Response · Authors · 2021-11-18
> **Response to Reviewer cqfk(Part 4)**
>
>
>   |                          | 50% sym   | 90%sym    | 40% asym | 60% total with 50% open-set noise |
> |--------------------------|-----------|-----------|----------|-----------------------------------|
> | S3 selection + S3 training | **96.30** | **95.20** | **96.0** | **94.81** |
> | S3(Moit train)           | 95.27     | 69.52     | 94.75    | 94.33                             |
> | S3(Loss-based selection) | 95.61     | 46.52     | 95.04    | 94.64                             |
> | Moit train + Loss-based selection |   95.75   |    10  |   91.67  |       80.41                 |
> *Table Q2.3.2: Additional ablations*
>
> > **Q2.4** ‘’From Figure 2 (a), the network still has good performance when $\theta_r$ approximates 1. Does it mean that relabelling is not very necessary? I think it will be much better if authors calculate the F-score to represent the quality of sample selection and relabelling especially when open-set label noise involved.''
>
> Thank you for the comment. For lower noise ratio, the relabeling gave smaller gains. However, much larger gains were observed in the case of higher noise ratio (Fig.2 in original paper). As we can see, the higher the noise ratio, the more important sample relabelling is. Unfortunately, we are not sure what the Reviewer meant with the F-score experiment. Should the reviewer provide additional clarification, we'd be happy to conduct the experiment and produce the requested F-score.

---

> ### Author Response · Authors · 2021-11-29
> **Reply to update**
>
> Thank you for your time to go through our response so promptly. However, there seems to be an misunderstanding. An apple to apple comparison between our method and C2D is not possible as we use the same hyperparameters for all experiments to show the high robustness of S3 in a dataset-agnostic style. C2D use different hyperparameters per experiment. As revealed in Fig.3(a), it is clear and reasonable that a lower $\theta_r$ can help the model relabel more samples and thus include more samples in the training. By simply changing a single parameter ($\theta_r$) per dataset we significantly outperform C2D.
>
> |     | 90% CIFAR100 | 95% CIFAR100 |
> |-----|--------------|--------------|
> | C2D | 58.70%       | 37.39%       |
> | S3  | 60.83%       | 54.14%       |
>
> We will also upload our code online later and all experiments can be validated.

---

> > ### Comment · Reviewer_cqfk · 2021-11-29
> > **Sorry for misunderstanding**
> >
> > I apologize to the authors that I did not notice the different experiment settings between S3 and C2D.
> > It seems that the performance of S3 does outperform C2D for extreme noise rate under fair comparison. I strongly suggest the authors
> > add this part of the experiment since it will make S3 more appealing.
> >
> > Thanks for releasing the code. It will be very beneficial for the community.
> >
> > Finally, it is worth noting that C2D is only one example. There are also other works that, similar to C2D, adopt contrastive pre-training for learning with noisy labels. For example, this work (https://arxiv.org/pdf/2108.09154.pdf) also achieves very high performance under extreme noise rate with [contrastive pre-training + CoL finetuning].  It will be better if the authors add more relevant references, or possible fair comparisons in the revised submission.
> >
> > Unfortunately, since my other major concerns still remain, I decide to keep my original score.

---

### Official Review · Reviewer_bqh4 · 2021-11-01

**Correctness:** 3
**Technical Novelty And Significance:** 3
**Empirical Novelty And Significance:** 3
**Recommendation:** 6
**Confidence:** 4

**Main Review:**

**Strength**
- State-of-the art results in different settings.
- The methods works both with open- and closed-set noise.
- While the ideas of neighborhood-based sample selection, sample relabeling, and consistency regularization are well-known, the particular implementation is novel and interesting.
**Weaknesses**
- The presentation should be improved. The authors explain well-known concepts like softmax gradient but some details of the proposed methods are explained briefly or not at all.
- There is a number of existing methods that you don't compare to: C2D (Zheltonozhskii et al., 2021), which you cite but do not include in comparison and which outperforms you at least in some setups (e.g., CIFAR-100 90% noise -- 58.45±0.30% as opposed to 56.8% you report); FaMUS (Xu et al., 2021). Similarly AugDesc (Nishi et al., 2021) is mention in Tab. 1 but not in main comparison.
**Comments and questions**
- Most of existing papers report top-1 and top-5 accuracy for WebVision training, both on ImageNet and WebVision validation sets. Can you provide those numbers?
- Comparison with other self-consistency approaches could be helpful (e.g., L2 loss is often used in semi-supervised learning or contrastive loss)
- Different parameters are compared on different datasets, which makes the comparison hard. You should use same settings (or multiple settings, but for every experiment). Also, you show dependence on $\theta_r$ but not $\theta_s$. Why?
- Using average of at least two runs is better than nothing, but in some cases differences are clearly not significant, e.g. Tab. 1 and Tab. 3. Showing standard deviation would make clear which improvements are significant.
- Probably balancing would be more effective in unbalanced settings, either synthetic or real-life (maybe Clothing1M, Xiao et al., 2015)?
- For ablation study, what happens if you use only one of sample selection or sample relabeling?
- Tab. 3: balanced selection refers to weighting $p'$ by $\pi^{-1}$?


**Minor comments and typos**
- Section 3.3 "are the inverses of the entries of the vector $\pi$ of the class probabilities in the whole dataset" is hard to read. I assume you meant something like "The vector $\pi$ contains the class probabilities in the whole dataset, and we denote by $\pi^{-1}$ its elementwise inverse".
- Section 4.4 has wrong citation format. It should be "DivideMix (Li et al., 2020a)" and not "DivideMix Li et al., (2020a)" (there are more of those in this paragraph and possibly elsewhere).
- Section 4.2 "Genrally->Generally"
- Table 7 caption "ANIMAl->ANIMAL"


**Post-rebuttal/revision**
The authors have addressed many concerns, especially experiment-wise. New comparisons were added (C2D still missing from Tab. 6 though -- the results for WebVision appear in table 4 in the paper), as well as Clothing1M experiments. The presentation still can be improved, in particular, I suggest to proofread the newly-written part.

**References**

Nishi, Kento, et al. "Augmentation strategies for learning with noisy labels." Proceedings of the IEEE/CVF Conference on Computer Vision and Pattern Recognition. 2021.

Xiao, Tong, et al. "Learning from massive noisy labeled data for image classification." Proceedings of the IEEE conference on computer vision and pattern recognition. 2015.

Xu, Youjiang, et al. "Faster meta update strategy for noise-robust deep learning." Proceedings of the IEEE/CVF Conference on Computer Vision and Pattern Recognition. 2021.

Zheltonozhskii, Evgenii, et al. "Contrast to Divide: Self-Supervised Pre-Training for Learning with Noisy Labels." arXiv preprint arXiv:2103.13646 (2021).

**Summary Of The Paper:**

The paper proposes a two-stage approach to learning with noisy labels (LNL).
1. a. Clean sample selection based on cosine similarity with k nearest neighbors in embedding space: the average of class distribution of those neighbors should be consistent with the label for sample to be selected.
    b. Noisy sample relabeling using temporal self-ensemble: the prediction is defined is average over last L epochs. The sample is relabeled if the confidence of the prediction is higher than some threshold.
2. Training with self-consistency regularization: the regular training with mixup regularization is performed on selected and relabeled samples, with consistency regularization in form of cosine similarity.
The paper tests the performance of method on multiple datasets both with synthetic and real-life noise.

**Summary Of The Review:**

The proposed method is novel and interesting and shows promising results in various settings.  My main problem is quality of writing and presentation. if the authors can address my comments and questions and improve the general presentation quality, I'll happily increase the recommendation to accept.

**Post-rebuttal/revision**
The authors have done good job addressing my concerns (as well as other reviewers'). The experimental results look interesting and while presentation (in particular writing) can be improved, I think the paper should be accepted.

---

> ### Author Response · Authors · 2021-11-18
> **Response to Reviewer bqh4(Part 1)**
>
> > **Q1.1** ``The presentation should be improved. The authors explain well-known concepts like softmax gradient but some details of the proposed methods are explained briefly or not at all.''
>
>  Thank you for the comment. We will improve the presentation by reducing some of the less relevant material and by expanding on the details and adding experimental results that give insight. We will update the result tables to compare with the methods that the reviewers mention (Tables.4/6 in original paper), and include a discussion on why the selection, relabelling and training scheme that we adopt perform better than other methods in the literature; this discussion is supported by experimental evidence as well (see Table Q2.1.2 in the response to Reviewer cqfk Q2.1).
>
> >  **Q1.2**  ``There is a number of existing methods that you don't compare to: C2D (Zheltonozhskii et al., 2021), which you cite but do not include in comparison and which outperforms you at least in some setups (e.g., CIFAR-100 90\% noise -- 58.45±0.30\% as opposed to 56.8\% you report); FaMUS (Xu et al., 2021). Similarly AugDesc (Nishi et al., 2021) is mention in Tab. 1 but not in main comparison.''
>
> Thank you for the comment, we did the comparisons (please see Table Q1.2, Table Q1.3 Table Q1.7). We will include these comparisons in the paper. Please note that FaMUS applied different noise ratio settings, however our method achieved higher accuracy with even higher noise.
>
> | Noise Dataset | 20% sym (CIFAR10) | 40% sym (CIFAR10) | 50% sym (CIFAR10) | 60% sym (CIFAR10) | 80% sym (CIFAR10) | 90% sym (CIFAR10) | 40% asym (CIFAR10) | 20% sym (CIFAR100) | 40% sym (CIFAR100) | 50% sym (CIFAR100) | 60% sym (CIFAR100) | 80% sym (CIFAR100) | 90% sym (CIFAR100) |
> |---------------|-------------------|-------------------|-------------------|-------------------|-------------------|-------------------|--------------------|--------------------|--------------------|--------------------|--------------------|--------------------|--------------------|
> | AugDesc       | 96.3              | /                 | 95.4              | /                 | 93.8              | 91.9              | 94.6               | 79.5               | /                  | **77.2** | /                  | 66.4               | 41.2               |
> | C2D           | 96.4              | /                 | 95.3              | /                 | 94.4              | 93.6              | 93.5               | 78.7               | /                  | 76.4               | /                  | 67.8               | **58.7** |
> | FaMUS         | /                 | 95.4              | /                 | 95.0              | /                 | /                 | /                  | /                  | 75.9               | /                  | 73.6               | /                  | /                  |
> | S3            | **96.6** | /             | **96.3** | /                 | **95.7** | **94.9** | **96.0** | **79.7** | /                  | **77.2** | /                  | **70.4** | 56.8               |
>
> *Table Q1.2 Comparisons to FaMUS, C2D and AugDesc*
>
>  There are a couple of cases in which other methods perform slightly better, but overall it is clear what we outperform all the methods you mentioned, in several cases, by large margins. We re-iterate the simplicity of our method that uses off-the-shelf training methods (section 3.4) without bells and whistles like co-training and ensembles, and a selection and relabelling strategy that can be expressed in 3 simple equations (section 3.3). Please also note that C2D is interesting but is not peer reviewed (Arxiv).

---

> > ### Comment · Reviewer_bqh4 · 2021-11-20
> > **Short answer**
> >
> > **Q1.1**
> > As I said, this is my major concern, so I'll read carefully the revised version of the manuscript when you upload it before making final decision.
> >
> > **Q1.2**
> > Thanks for update

---

> > > ### Author Response · Authors · 2021-11-21
> > > **Further response**
> > >
> > > Thanks for your reply.
> > >
> > > > **Q1.1**  As I said, this is my major concern, so I'll read carefully the revised version of the manuscript when you upload it before making final decision.
> > > >
> > > Thank you so much for you time, please check the uploaded revised paper.
> > >
> > > >**Q1.3** Note that you're missing FaMUS and C2D in this comparison too.
> > >
> > > Thanks for your reminder. Now FaMUS results are included in the updated reply and the revised paper, while C2D didn't report results on WebVision and ILSVRC2012 dataset.
> > >
> > > > **Q1.4**  What I meant is to replace the consistency measure (cosine similarity), to, e.g., L2 within your method as opposed to changing the whole scheme.
> > >
> > > Thanks for the clarification,  please check the new Appendix D for self-consistency with L2 loss.

---

> > > > ### Comment · Reviewer_bqh4 · 2021-11-30
> > > > **Answer**
> > > >
> > > > C2D includes the results for webvision (table 4 in the paper).
> > > > The revised paper improved clarity, though I suggest to proofread the newly-written parts for grammar and typos.
> > > > I've updated my score accordingly.

---

> ### Author Response · Authors · 2021-11-18
> **Response to Reviewer bqh4(Part 2)**
>
> > **Q1.3**  ``Most of existing papers report top-1 and top-5 accuracy for WebVision training, both on ImageNet and WebVision validation sets. Can you provide those numbers?''
>
> Thank you for the comment. We will include the top-1 and top-5 accuracy in the paper -- please see Table Q1.3 below. For the top-5 accuracy results, our method performs the best.
>
> |    Method                          |  WebVision(Top1)   |  WebVision(Top5)   |  ILSVRC2012(Top1)    |   ILSVRC2012(Top5)   |
> |------------------------------|-----------|-----------|------------|-----------|
> | DivideMix(InceptionResNetV2) |   77.32   |   91.64   | **75.20** |   90.84   |
> | ELR(InceptionResNetV2)       |   76.26   |   91.26   |    68.71   |   87.84   |
> | ELR+(InceptionResNetV2)      |   77.78   |   91.68   |    70.29   |   89.76   |
> | NGC(InceptionResNetV2)       |   79.16   |   91.84   |    74.44   |   91.04   |
> | RRL(InceptionResNetV2)       |    76.3   |    91.5   |    73.3    |    91.2   |
> | GJS(ResNet50)                |   77.99   |   90.62   |    74.33   |   90.33   |
> | DivideMix(resnet18)          |   76.08   |     /     |      /     |     /     |
> | ELR(resnet18)                |   73.00   |     /     |      /     |     /     |
> | MOIT+(resnet18)              |   78.76   |     /     |      /     |     /     |
> | S3(resnet18)                 | **80.12** | **92.80** |    74.84   | **91.26** |
> *Table Q1.3 WebVision with more results*
>
> > **Q1.4** ``Comparison with other self-consistency approaches could be helpful (e.g., L2 loss is often used in semi-supervised learning or contrastive loss)''
>
> Several of the methods to which we compare, do use consistency-based regularizations. For example, C2D uses this in a pretraining stage, while MOIT [1] and RRL [2] applied supervised contrastive loss. Please see Tab.4 and Tab.6 in the original submitted paper [RRL was named as ProtoMix]. We would be happy to clarify further if needed.
>
> *[1]  Diego Ortego, Eric Arazo, Paul Albert, Noel E O’Connor, and Kevin McGuinness. Multi-objective interpolation training for robustness to label noise. In Proceedings of the IEEE/CVF Conference on Computer Vision and Pattern Recognition, pp. 6606–6615, 2021.*
>
> *[2] Li, Junnan, Caiming Xiong, and Steven CH Hoi. "Learning From Noisy Data With Robust Representation Learning." _Proceedings of the IEEE/CVF International Conference on Computer Vision_. 2021.*
>
> > **Q1.5**     ``Different parameters are compared on different datasets, which makes the comparison hard. You should use same settings (or multiple settings, but for every experiment). Also, you show dependence on $\theta_r$ but not $\theta_s$. Why?''
>
>  We used the same hyper-parameters for all CIFAR experiments (namely $\theta_s=1$, $k=200$ and $L=10$) except in specific ablations for the corresponding hyper-parameter. For different noise ratios, different $\theta_r$ are optimal, but the ablations (Fig. 2 in the original submitted paper) shows that we surpass the state of the art for several configurations of them.
>
> Regarding $\theta_s$:  we fixed $\theta_s = 1$ for all experiments. For your perusal, we list below results with $\theta_s = \{0,0.8,1\}$ in Table Q1.5 below -- those will be included in the paper as well.
>
> | $\theta_s$ | 50% sym   | 90%sym    | 40% asym | 60% total with 50% open-set noise|
> |------------|-----------|-----------|----------|-------------------------------|
> | 0          | 77.35     | 33.64     | 87.95    | 78.69                         |
> | 0.8        | 96.11     | 95.01     | 95.35    | **94.82** |
> | 1          | **96.30** | **95.20** | **96.0** | 94.81                         |
> *Table Q1.5: Accuracy with respect to different values of $\theta_s$*

---

> > ### Comment · Reviewer_bqh4 · 2021-11-20
> > **Thanks for the reply**
> >
> > **Q1.3**
> >
> > Note that you're missing FaMUS and C2D in this comparison too.
> >
> > **Q1.4**
> > What I meant is to replace the consistency measure (cosine similarity), to, e.g., L2 within your method as opposed to changing the whole scheme.
> >
> > **Q1.5**
> > Thanks for clarification.

---

> ### Author Response · Authors · 2021-11-18
> **Response to Reviewer bqh4(Part 3)**
>
> > **Q1.6** ``Using average of at least two runs is better than nothing, but in some cases differences are clearly not significant, e.g. Tab. 1 and Tab. 3. Showing standard deviation would make clear which improvements are significant.''
>
> Thank you for the comment. In Tab.1 of the original submission, we wanted to show the effect of different augmentation combinations in our method. Our results are consistent with the findings of AugDesc [3] that one should use weaker augmentations for sample selection and relabeling, and strong augmentations for training. This is consistent in all settings and we feel there is sufficient evidence to support this conclusion without additional experiments. We also found that the higher the noise ratio, the more prominent the differences are with different augmentation strategies. We will explicitly comment on that in the revised paper.
> In Tab.3 of original paper, we examine the effect of balancing. We agree that the differences are smaller in comparison to other design choices we make. Following the reviewer's suggestion (see Q1.7), we experimented on Clothing1M and show that balancing does help -- the results are summarised in Table Q1.7, and will also be included in the updated paper.
>
> *[3] Nishi, Kento, et al. "Augmentation strategies for learning with noisy labels." _Proceedings of the IEEE/CVF Conference on Computer Vision and Pattern Recognition_. 2021.*
>
> > **Q1.7**     ``Probably balancing would be more effective in unbalanced settings, either synthetic or real-life (maybe Clothing1M, Xiao et al., 2015)?''
>
> Thank you for your suggestion to test balancing on Clothing1M dataset. We did so and the results below (see Table Q1.7) show that balancing does help, and S3 get comparable results to SOTA with simpler model structure. We will include the results in the paper as well.
>
> |  Method                | Top-1 ACC |
> |------------------|-----------|
> | DivideMix        | 74.76     |
> | C2D              | 74.30     |
> | FaMUS            | 74.40     |
> | Augdesc          | 75.11     |
> | ELR              | 72.87     |
> | ELR+             | 74.81     |
> | PDLC             | 74.15     |
> | RRL              | 74.84     |
> | S3 W/O Balancing | 74.12     |
> | S3               | 74.83     |
> *Table Q1.7: Clothing1M results*
> >**Q1.8** "For ablation study, what happens if you use only one of sample selection or sample relabeling?"
>
> Thank you for the comment. In Fig. 2 of original paper, we had reported results with sample selection and different thresholds for relabeling, including $\theta_r = 1$ which corresponds to no relabeling. Following the reviewer's suggestion we do additional experiments with different thresholds for sample selection, including $\theta_s = 0$ which corresponds to no sample selection (i.e., all samples considered as clean subset). The results can be found in Table Q1.5. It is clear that to obtain high accuracy both steps, as proposed in our paper, are needed.
>
> >**Q1.9**      "Tab. 3: balanced selection refers to weighting $p'$ by $\pi^{-1}$?"
>
> Yes.
> >**Q1.10** Minor comments and typos
>
> Thank you for your careful proofreading, we will modify all the typos in the updated version of paper.
> >**Q1.11**  ''The proposed method is novel and interesting and shows promising results in various settings. My main problem is quality of writing and presentation. if the authors can address my comments and questions and improve the general presentation quality, I'll happily increase the recommendation to accept.''
>
> Thank you very much for the detailed review. We have tried to address all the comments including comparisons to all the works that the reviewer mentions, both experimentally and methodologically. We will also improve the presentation in the revised paper by including all the extra experiments and the discussion. Please, see the responses to the other reviewers about additional material that will be included.

---

### Author Response · Authors · 2021-11-18
**General response to the reviewers and AC**

We thank the reviewers for their critical assessment of our work.
In the following we address their concerns point by point. We will include new material based on the reviewers' comments and we will update/expand the experimental results in the revised paper. In the responses below we include the updated tables and new results. We are working on updating the paper with the new material and we will post here when we will have that ready.

---

### Author Response · Authors · 2021-11-21
**Thanks again to the reviewers and AC, we have uploaded the revised paper**

Once again, we thank all reviewers for their comments. We have uploaded the update version of the paper to cater for all the points that the reviewers raised. Please, find the changes with blue colour in the new manuscript. To summarise:

-  We have updated the introduction, so as to better position our work in the related literature and highlight the differences with other works and their significance.
-  We have added short introductions to sections 3 and 4.
- We have removed the trivial derivation of the gradient of the cross entropy and moved implementation details and the augmentation ablations in Appendix.
- We have added new ablations studies to a) show the performance of the sample selection and relabeling (section 4.2.1, page 7) and b) show the significance of each individual component (section 4.2.3, page 8).
-  An Appendix where we investigate on the influence of L2 as a self-consistency loss (Appendix D).

If the reviewers have any further questions, we are happy to discuss and answer to them.

---

### Public Comment · ~Chen_Feng3 · 2023-03-03
**Updated version available!**

Citation: @inproceedings{Feng_2022_BMVC,
author    = {Chen Feng and Georgios Tzimiropoulos and Ioannis Patras},
title     = {SSR: An Efficient and Robust Framework for Learning with Unknown Label Noise},
booktitle = {33rd British Machine Vision Conference 2022, {BMVC} 2022, London, UK, November 21-24, 2022},
publisher = {{BMVA} Press},
year      = {2022},
url       = {https://bmvc2022.mpi-inf.mpg.de/0372.pdf}
}

Code repo: https://github.com/MrChenFeng/SSR_BMVC2022

---

### Decision · Program_Chairs · 2022-01-20

**Decision:**

Reject

**Comment:**

This work describes a two-stage method for learning with noisy labels. The crux of the reviews, discussions with the authors and post-rebuttal discussions between reviews (and myself) was related to the novelty of this work. The main concern is that while this body of work presents a relatively solid method (from an empirical point of view), the underlying components are not altogether that novel, and have been used in the context of learning with noisy labels before. Fundamentally, the proposed S3 method did not feel *convincingly* better, given its relative lack of novel technical insights. I appreciate that this is a frustrating reasoning to get -- after all, much of what we do in empirical ML is combinations of existing things. Ultimately, there was consensus amongst the reviewers that the work did not have sufficient insights or such outstanding empirical results so as to overcome this relative lack of technical novelty.

All the reviewers have engaged meaningfully in discussions, provided constructive feedback and I hope that this will make subsequent iterations of this work better in many dimensions.